# Offline Calibration for Infant Gaze and Head Tracking across a Wide Horizontal Visual Field

**DOI:** 10.3390/s23020972

**Published:** 2023-01-14

**Authors:** Chiara Capparini, Michelle P. S. To, Clément Dardenne, Vincent M. Reid

**Affiliations:** 1Center for Research in Cognition & Neuroscience (CRCN), Université Libre de Bruxelles, 1050 Brussels, Belgium; 2Department of Psychology, Lancaster University, Lancaster LA1 4YF, UK; 3Research and AI Labs, Smart Eye AB, 413 27 Gothenburg, Sweden; 4School of Psychology, University of Waikato, Hamilton 3240, New Zealand

**Keywords:** eye tracking, calibration, gaze, vision, visual field, infancy, developmental psychology

## Abstract

Most well-established eye-tracking research paradigms adopt remote systems, which typically feature regular flat screens of limited width. Limitations of current eye-tracking methods over a wide area include calibration, the significant loss of data due to head movements, and the reduction of data quality over the course of an experimental session. Here, we introduced a novel method of tracking gaze and head movements that combines the possibility of investigating a wide field of view and an offline calibration procedure to enhance the accuracy of measurements. A 4-camera Smart Eye Pro system was adapted for infant research to detect gaze movements across 126° of the horizontal meridian. To accurately track this visual area, an online system calibration was combined with a new offline gaze calibration procedure. Results revealed that the proposed system successfully tracked infants’ head and gaze beyond the average screen size. The implementation of an offline calibration procedure improved the validity and spatial accuracy of measures by correcting a systematic top-right error (1.38° mean horizontal error and 1.46° mean vertical error). This approach could be critical for deriving accurate physiological measures from the eye and represents a substantial methodological advance for tracking looking behaviour across both central and peripheral regions. The offline calibration is particularly useful for work with developing populations, such as infants, and for people who may have difficulties in following instructions.

## 1. Introduction

Eye tracking represents an accessible and non-invasive tool capable of measuring looking times, recording saccadic behaviours, assessing physiological ocular measures, such as pupil dilation, and shedding light on mental processes [1,2,3,4,5]. Due to the low level of cooperation required from participants, eye-tracking techniques are particularly suitable for non-verbal participants and participants who are unable of following instructions, such as infants. In recent decades, eye trackers have allowed researchers to explore infant development and gain significant insights into early perception and cognition [6,7,8]. Even though eye tracking is a valid tool for addressing a variety of research questions about infant perception and cognition, researchers are constantly faced with technical challenges and constraints. As outlined by Oakes [8], some of these challenges include tracking head movements (raising the possibility of missing data if participants move the head outside the trackable area and the system must find the eye coordinates again), obtaining a good calibration, and establishing the experimental design implementation and data processing required to analyse the data. Whereas the two latter aspects are more related to the analytical and technical skills of the researcher before and after data acquisition, the former two are closely linked with the infant population and as such will be the focus of this paper. Additionally, some challenges are related to the physiology of the infant eye, both in terms of anatomical differences between the structure of the developing eye compared with the adult eye, and in terms of the typical features of infants, such as the tendency for wet eyes. Anatomical differences will be also considered in the present work.

To some extent, the impact of head movements depends on the trackable area available for eye tracking which, in turn, depends on the device used to measure eye movements [9]. Two broad categories of eye-tracking devices are generally available for infant participants: remote and head-mounted (or wearable) systems. The most well-established eye-tracking paradigms in infancy have taken advantage of non-intrusive remote eye-tracking systems (e.g., [10,11,12]). Most commercial solutions support regular flat screens up to 27 inches in size (or up to 30 inches with some limitations). Although small head movements are tolerated by many eye-tracking solutions, the eye’s image is usually lost outside the screen area and has to be reacquired following each tracking loss. According to Tomalski and Malinowska-Korczak [13], infant participants spend about 10% of a standard eye-tracking session looking away from the monitor, resulting in missed data each time the system has to recover the eye’s image. These spatial constraints, restricted by the area of a regular screen, mean that gaze-orienting behaviours are mostly investigated through the contribution of the eyes while the head is held in a relatively stable position. In contrast, in everyday situations we orient thanks to both eyes and head movements. Even though such infant paradigms using regular screens have enabled valuable insights into infant visual behaviour in response to stimuli, more ecologically valid investigations that span across wider visual locations have been limited. To the best of our knowledge, so far, only one study by Pratesi and collaborators [14] has adopted a remote eye-tracker to investigate infant gaze behaviour beyond near-peripheral locations (+/−30°). This was achieved by using five screens (a central screen and two additional screens on each side) across a 120° field of view. In addition to remote eye-tracking systems, the development of head-mounted eye-tracking systems has enabled an alternative method that allows free head and body movements and the possibility of investigating a wider three-dimensional space [15]. Some young participants do not, however, respond positively to a wearable system on their heads, and may easily displace or remove the device. Further, these devices can be complicated to set up, resulting in higher overall attrition rates [16].

The current study is aimed at testing a remote system that is not invasive for the participant and, at the same time, measures across a wide field of view of 126°. We defined a wide field of view extending beyond the near-peripheral locations which can be investigated using regular screens supported by most remote eye-tracking solutions. Measuring gaze movements across the visual field opens the possibility of studying the developing visual behaviour in a more naturalistic and unconstrained visual environment. The limitation of a restricted trackable area, in which the pupil position can be accurately detected, is overcome by using multiple infrared cameras. In the present work, a 4-camera system allows tracking of both the contributions of the eye relative to the head and the contributions of the head relative to the spatial environment. This work builds upon the initial investigation across a wide visual field of Pratesi and colleagues [14], who piloted a similar system on a small group of nine infant participants. The present work extends this approach to a larger sample and to a single and wider screen, while taking advantage of a new software specifically adapted to the developing head and eyes. The current multi-camera setup enables researchers to investigate infant perception and cognition beyond standard screen sizes and, potentially, to define a tracking area even without a screen (see, for instance, applications of similar eye-tracking systems in the automotive field in order to track drivers’ eye movements across different car spaces; e.g., [17]). The applications of this system include a range of studies investigating visual behaviour beyond a limited trackable area in the context of a participant who is less constrained to direct their visual attention to a standard screen space. Visual orientation could be monitored while participants move their heads in an active ‘real-world’ exploration. At the moment, similar investigations are mostly carried out with head-mounted systems, with the limitations described above.

The second challenge of eye tracking in infancy research which is addressed in this paper is calibration. Every eye-tracking system relies on calibration and quality of the data often depends on this [7]. In fact, the data provided by the system (e.g., gaze positions) must be mapped onto the stimulus/display area. Eye-tracking data collected from infants are not always reliable and this can even lead to apparent differences in gaze behaviour when different groups of individuals are compared [18,19]. Among the relevant parameters in evaluating data quality, this paper specifically focuses on (1) spatial accuracy, as this is limited by the quality of the calibration procedure, and (2) robustness (i.e., data loss), as it is linked with the trackable area available. For a focus on precision, a third parameter of data quality, see Wass et al. [19], who investigated the variables that low precision may influence when tracking the infant’s gaze on standard displays. Spatial accuracy (offset) refers to the distance between the actual location of the stimulus that a participant is looking at relative to the gaze points recorded/extracted by the eye-tracking system [5,20]. Traditionally, a good spatial accuracy is achieved by asking participants to maintain fixation on a number of small visual targets (usually 9 for adults) at predefined locations on the screen. This means calibration is more difficult with young infants who cannot follow such instructions, resulting in a spatial offset of 1–2° [1]. In developmental studies, highly attractive stimuli (e.g., moving or looming colourful images paired with sounds) are typically used and calibration points are significantly reduced to 5 or even 2 in some cases [6]. Notably, not all attractive stimuli result in a high-accuracy calibration. A recent investigation [21] compared the impact of different calibration targets on infants’ attention and found that some targets, such as complex concentric animations or stimuli with the highest contrast at their centre, elicited more accurate gaze than others. In addition, taking the infant’s limited attention span into account, calibration should ideally be as brief as possible so that the infant is not too tired and remains cooperative during the following experimental procedure [6]. For these reasons, optimal infant gaze calibration is not always achievable before the start of an experiment and, as it stands, there are currently no standard or prescribed calibration guidelines for researchers (see [22] about publishing eye-tracking data in infancy research). For instance, important considerations, such as the criteria that determine whether the calibration is valid and whether it should be adjusted or repeated during the experiment, are not standardised across studies [8]. The efficacy of a calibration procedure in producing accurate gaze measurements has been rarely included in empirical infant research, although it has been previously recommended as a factor of importance for methodological descriptions [18,23]. Studies using young participants have revealed evidence of systematic calibration errors and low spatial accuracy compared with the manufacturer’s estimates [18,24,25].

More generally, eye trackers often show a systematic error, even with adult data and after careful online calibration [26]. To overcome this issue, post hoc (or implicit) offline calibration has been proposed as a successful approach to replace calibration methods that require explicit collaboration from the participants [27,28] or as an additional step to improve data quality [24,26,29,30,31]. This procedure normally includes recalibrating individual gaze points at various times during the study by correcting the error between the recorded gaze data of a participant and the actual location of the visual stimulus. To date, offline calibration methods of correct eye-tracking offsets have rarely been adopted in infancy research [24]. In the present work, we combined an online system calibration with a novel offline implicit gaze calibration to improve the spatial accuracy of the eye-tracking system. The latter was possible as visual targets appeared at stable and predetermined positions during the experiment.

## 2. Materials and Methods

### 2.1. Dataset

Eye-tracking data from an ongoing project which involved 35 (18 females) 9-month-old participants were used for this study. Four infants were excluded either for technical issues which caused complete data loss (*n* = 3) or for the unintended inclusion of parental gaze (*n* = 1). Thirty-one infants (16 females) with a mean age of 275 days (SD = 9.1 days) constituted the final sample. All infants were born at full term (>37 weeks) with normal birth weight (2.5–4.5 kg) and were typically developing. Sensory impairments and eye infections were among the criteria that excluded participation in this study. Participants were recruited via email or phone invitation from the Lancaster University’s Babylab database of volunteer families. All parents gave informed written consent prior to the beginning of the study. Families received a £10 travel compensation and a storybook to thank them for participating. The Faculty of Science and Technology Research Ethics Committee of Lancaster University reviewed and approved the protocol of the study (project ethics approval reference no. FST19121, under the programme of studies with approval reference no. FST18067). This work was conducted according to the principles expressed in the Declaration of Helsinki.

### 2.2. Apparatus

Eye-tracking data were recorded with a Smart Eye Pro 4-camera system (Smart Eye AB, Gothenburg, Sweden) running at 60 Hz. This is a corneal reflection remote system capable of recording gaze at 0.5° accuracy under ideal conditions. A machine learning algorithm initially detects the participant’s facial features. Then, the system tracks the centre of the pupil together with glints (i.e., the reflections of the infrared flashes on the cornea) to find the centre of the eyes [14]. For this project, Smart Eye provided a modified version of the Smart Eye Pro software (Gothenburg, Sweden), v8.2, which was specifically adapted to the anatomy of the developing head and eyes and enabled us to maximise the performance of the system with infant participants. In fact, earlier software versions were based on face recognition algorithms developed on adults’ facial features and proportions that were less reliable with young participants. These modifications of the head model have been recently released as an additional child head module that can be enabled in Smart Eye Pro from version 9.0 onwards. Smart Eye Pro features flexible camera placements that can be adjusted based on the needs of the user. The number of cameras used can vary depending on the span of the visual environment being investigated. In order to cover the entire horizontal field of view (FOV) of 126°, four cameras were positioned below the display monitor, a 49-inch Samsung LC49HG90DMM curved screen (120.30 cm width and 52.55 cm height without stand, 3840 × 1080 pixels resolution). Cameras were equidistantly placed on a 3-arm flexible support located just below the screen, two on the left- and two on the right-hand side of the participant. Depending on head position relative to the screen, the cameras captured data from both eyes or from a single eye and helped to account for large head movements. Three 850 nm wavelength infrared flash producers were placed along the camera support, one in between the two central cameras (by the screen midline) and the other two by the edges of the screen (see Figure 1a,b). Using active infrared illumination to illuminate the participant’s face, the system is described by the manufacturer as ambient light-independent. As such, the effect of ambient lighting conditions should not necessarily affect data outcomes. This system enabled an accurate identification of gaze direction in a wide visual field by measuring infants’ eye and head components and by using this information to extract their overall orienting behaviours. Visual stimuli were displayed using the Psychophysics Toolbox extensions, version 3 [32,33] in MATLAB, version 2018a (MathWorks, Natick, MA, USA), which was running on a Dell Latitude 5491 computer managed by the experimenter. The visual stimulus presentation, managed by MATLAB and the Smart Eye eye-tracking recordings, was connected via the User Datagram Protocol (UDP) and both were reading the local time from the computer.

### 2.3. Experimental Procedure

Infant participants sat on their caregivers’ lap at 40 cm from the centre of the curved screen. At this distance, the screen covered 126° horizontal FOV and enabled the presentation of visual stimuli at 60° from either side from the centre of the monitor. Notably, this approximately covered the full visual field available to infants at the tested age [34,35]. While most commercial eye-tracking solutions work best at a specific distance (usually between 55–70 cm), the cameras adopted here have an optimal camera–eye distance, ranging from 30 to 300 cm due to adjustable lenses and camera positioning (as claimed by the manufacturer), and thus allow for more flexible setups. Caregivers were instructed to maintain their infant in a stable, upright position at a constant distance from the screen. Seating on the caregiver’s lap helped in reducing the infant movements (see [13] for a comparison of seating arrangements on infant behavior during eye-tracking studies), but a similar procedure with a Smart Eye system has been also implemented using an infant chair [14]. Caregivers were also requested to avoid talking and interfering with the infant’s looking behaviour during the experimental procedure. Lights were switched off, but the room was lit by the computer screen (screen luminance during the experimental procedure was around 25–26 cd/m^2^). This lighting choice limited the possibility of distractions and ensured constant lighting across participants.

Before the experiment started, an eye-tracking system calibration was performed in two phases. First, the positions of the four cameras were examined to ensure that the infant’s face was well centred on all cameras (if not, each camera could be rearranged slightly until the participant’s face fell within a central headbox displayed on screen by the calibration software). The brightness (aperture) and focus settings of each camera were then visually assessed relative to the participant’s face and adjusted if required. Optimal aperture and focus settings were reached when the two bars surrounding the participant’s face in the Smart Eye Pro’s Graphical User Interface (GUI) approached their maximum capacity. In this phase, the caregiver was only instructed to maintain the infant in a stable position at the set distance from the screen so that the experimenter could quickly verify that the infant’s face was captured by all cameras. In our experimental setup, maintaining the same lighting conditions and the same participant positioning relative to the equipment across participants helped to reduce the adjustments of aperture, focus and camera placement to a minimum. After that, a small chessboard provided by the manufacturer was presented centrally (where the participant’s head will be located during the experiment) to allow the system to automatically calculate the current position of the four cameras with respect to the entire setup and, in turn, to extrapolate the head and eye positions. From this central position, the chessboard was tilted and rotated until the progress bar of the Smart Eye Pro’s GUI for each camera was filled, so that each camera could detect the chessboard. In this phase, the chessboard was required to be visible in each camera view. It is advisable to place it in front of the participant’s face, towards which the cameras are already facing, to obtain strong calibration results in that region. This step is required for the software to learn the positions and orientations of the cameras and should be performed each time the camera positions change. In our study, this procedure was performed before each testing session, as advised by the manufacturer. The system calibration was checked using a “Verify Camera Calibration” dialog where the experimenter could verify if calibration values were within parameters (labelled in green by the software interface) for all four cameras. If not, the system calibration was repeated. Following a successful system calibration, it is important that the positions and orientations of the cameras are not modified anymore. This system calibration was improved with an offline calibration procedure following the experimental session (see Section 2.5 Offline Calibration). A standard online gaze calibration was not possible given that our large display exceeded the limits of the infant’s visual field. Additionally, calibrating across the full screen area would have been too time consuming and would have reduced the infant’s cooperation for further data acquisition during the same experimental session.

Following the system calibration, the experiment could start. There was a total of 32 trials per infant. Using a gaze contingent eye-tracking procedure, each trial began with the presentation of a central attention grabber that disappeared as soon as the infant looked at the central area of the screen. The attention grabber was a blob stimulus with strongest contrast in its centre, presented within a Gaussian temporal envelope such that 100% contrast was attained in the middle of its total presentation time of 900 ms. The visual presentation was paired with a random synchronous tone. The audio-visual presentation of the central stimulus played continuously and was interrupted if the participant looked at it or if the participant did not look at the centre of the screen following the sixth repetition of the blob presentation. As soon as the central stimulus faded away, a peripheral target stimulus appeared in the left or right edge of the mid-periphery (+/−60°) in a randomised order and moved along the horizontal meridian towards the centre of the screen at 5°/s (covering 12 locations per side ranging from 60° to 32.5°, in 2.5° steps). Every 8 trials, the experimenter had to tap a key in order to advance the stimulus presentation. This was useful in case a break was needed or to readjust the participant’s position. Peripheral targets were faces taken from the Radboud Faces Database (see [36] for an example of the stimuli and for database validation). Each visual stimulus presented in this study covered an angular size of 5.88° at a 40 cm distance (180 × 180 pixels). Visual stimuli were presented on a uniform grey background. In this experimental procedure, the measures of interest were saccadic reaction times, used to detect the peripheral target and dwell times over the face regions. Saccadic reaction time was defined as the difference between the time in which the gaze reached the peripheral target area and the onset of the peripheral target. Dwell time was defined as the total time that the infant participant spent looking over the region of the detected moving face. The system sampling frequency of 60 Hz provided a good temporal resolution for these outcome measures. As outlined by Andersson and colleagues [37], with the adopted sampling frequency, reaction times can have an average sampling error of 8.33 ms (varying from 0 to 16.67 ms), whereas fixation durations are expected to have a mean sampling error centred on 0 (varying from −16.67 to 16.67 ms). Including system calibration, the entire eye-tracking session lasted on average 6–7 min. Data acquisition was performed by the same experienced eye-tracking operator for all infant participants.

### 2.4. Data Processing and Visualisation

First, a model of the three-dimensional curved screen environment (the so-called World Model) was defined within the Smart Eye Pro software in order to track gaze data relative to targets appearing on this particular display (Figure 1b). Smart Eye Pro computed the gaze intersections with the curved screen model in three dimensions. Raw data were exported from Smart Eye Pro as three-dimensional coordinates relative to the curved screen and were further processed with MATLAB. The parameters of interest used in the present work, as labelled and defined by the Smart Eye Pro software, included: Frame Information, Head Position, Raw Gaze, Filtered Gaze, and World Intersections. The parameter World Intersections distinguished the recorded timepoints in which the gaze intersected the screen, or world model, from those in which the gaze fell outside the screen. Due to the multi-camera system covering a wide visual area, raw gaze data were recorded even if the participant’ gaze was outside the screen. This minimised the amount of missing data. In fact, data were not lost throughout the experimental procedure when the gaze fell outside the monitor, as it is the case with other eye-tracking solutions. Raw data were later mapped and filtered relative to the screen area during the analysis using the World Intersections parameter. Robustness, calculated as the proportion of sample data in which the gaze location information was missing, is reported in the Results section.

Using simple trigonometric relations, the three-dimensional coordinates of the intersection between the screen and the gaze (*x*, *y*, *z*) were mapped onto the equivalent in pixels (u, v), so that gaze data and target position information were compatible. Gaze data obtained from Smart Eye and procedure data obtained from MATLAB were combined and synchronised with the real time clock. The individual gaze points were visualised together with the target locations to allow a visual check of data precision and accuracy. This visual inspection revealed a pattern of systematic error consistently towards the right side of the screen for the vast majority of participants. Since we were interested in saccadic reaction times in response to targets appearing in the left or right peripheral visual hemifields, reducing the systematic side offset and ensuring data accuracy was essential to drawing valid conclusions regarding the infants’ visual behaviour across the visual field. For this reason, an additional offline calibration step was included (see Section 2.5 Offline Calibration). Following this, central and peripheral areas of interest (AOI) were defined. The AOI were circular regions with a 180-pixel radius surrounding the central stimulus and the peripheral left and right areas where face stimuli appeared. Valid trials were defined as trials in which the participant’s gaze was within the central AOI at the offset of the central stimulus and the gaze reached a peripheral location after 100 ms and within 5 s from the onset of the peripheral stimulus. In order for a trial to be considered valid, gaze data could not be located outside the screen before the gaze reached the peripheral target AOI. Outliers were identified in deviating trials in which the latency to detect the peripheral target fell outside the interquartile range of latency across participants.

### 2.5. Offline Calibration

An interface was implemented in MATLAB to enable the experimenter to run an additional offline calibration procedure for the current project. This interface allowed us to visualise each participant’s gaze data together with the visual target on screen on a frame by frame basis. The locations of the visual targets displayed on screen during the experimental procedure were fixed (i.e., one central location and twelve peripheral locations per side) and were constantly marked on the interface. As the experiment progressed, the interface displayed both the stimulus on screen and the infant’s gaze data for any given time sample. Initially, the operator visualised the gaze data superimposed on the visual stimuli throughout the entire recording in order to identify the pattern of errors from trial to trial. On each trial, a single visual target appeared in the visual periphery and moved at fixed locations towards the centre of the monitor (at a constant velocity) along the horizontal meridian. Thus, the offset could be observed over an extended period of time, with one stimulus on screen at the time. After that, the experimenter could navigate back and forth through the data visualisation with steps of either 1, 10, or 100 frames. Once the experimenter identified a frame in which the gaze had reacted to the target change and gaze data appeared stable and clustered near the target location on the screen for at least 100 ms, they could tap a key to save both the current target position and the gaze coordinates of the participant. A total of 6 samples (3 samples relative to stimuli appearing on the left side and 3 samples relative to stimuli appearing on the right side) were chosen per participant. Only samples in which the visual target was on screen for an extended period of time (at least 1–2 s) were chosen; this occurred after every 8 trials (three experimental stages), when the target remained on screen until the experimenter made sure the participant was well positioned and attentive. A key press by the experimenter continued the presentation. Selection criteria for offline calibration samples included:Identifying the time segments in which a visual target was stable on screen (in this case, 3 intervals during the experimental procedure: after 8, 16 and 24 trials from the beginning of the test);Selecting suitable samples in which the gaze had reacted to the target change and gaze points were available, stable and clustered around the chosen target stimulus for at least 100 ms;Selecting 3 samples for each side of the screen following the above requirements.

This approach ensured that the 6 selected samples were collected from both sides of the screen/visual field, but were also representative of different experimental stages. The offset was calculated as the difference between current target position and gaze position. The average gaze offset was estimated from 6 individual offsets, described using horizontal and vertical coordinates (*x* and *y* axes, respectively) and its correction was added to the initial gaze points. Averaging across multiple individual offset coordinates relative to different locations and experimental stages reduced the possibility that the offline calibration procedure would distort the data, as could be the case choosing a single calibration point. In this phase, it is relevant to visualise the entire recording again, including the gaze data following offline calibration. The new corrected gaze coordinates were overlaid on the interface together with target positions and initial gaze coordinates in order to visually evaluate whether this corrective procedure was successful, i.e., to assess whether the new corrected gaze coordinates appeared closer to the targets throughout the entire experiment compared to the initial gaze coordinates (Figure 2).

## 3. Results

The system allowed tracking infants’ eyes movements on a wide visual area, covering mid-peripheral locations up to at least 60° either side. Due to head tracking, gaze data were not lost when the participants moved or turned their head. The robustness data revealed that, on average, 23.97% (SD = 11.93%) of the raw data were lost during the entire recording. The system could track data even beyond the screen area and approximately 28.00% of the data (SD = 10.14%) were recorded outside the screen. Data visualisation revealed that gaze data out the screen were often directed towards the flash producers just below the display monitor. Head tracking throughout the entire experiment allowed the monitoring of infants’ eye distance from the display. Although this distance was initially set to 40 cm, the variation throughout the recording was high, with median head distance values per trial ranging from 29.20 to 56.50 cm (M = 42.25 cm, SD = 4.73 cm). Importantly, the system could still accommodate this variation.

By combining information about target position and gaze position at selected time frames, it was possible to correct a mean offset of −42.22 pixels (SD = 38.88 pixels) on the *x*-axis and −44.81 pixels (SD = 53.61 pixels) on the *y*-axis. At a 40 cm distance, this corresponds to a −1.38° mean offset on the *x*-axis (SD = 1.27°) and a −1.46° mean offset on the *y*-axis (SD = 1.75°). At the individual level, the smallest average correction was −0.17 pixels (−0.01°) on the *x*-axis and −0.41 pixels (−0.01°) on the *y*-axis, whereas the largest average correction corresponded to 119.40 pixels (3.89°) on the *x*-axis and −207.47 pixels (−6.76°) on the *y*-axis (see Figure 3). This offline calibration procedure allowed the correction of an error that affected the majority of participants in the top-right direction (*n* = 27, 87.10%) and that could have contributed towards an incorrect data interpretation.

Further, mean horizontal and vertical offsets maintained a consistent direction across the three experimental stages during which the calibration points had been selected. Mean vertical offsets did not vary significantly across experimental stages, *X*^2^ _Friedman_ (2) = 2.52, *p* = 0.28. The experimental stage had a small effect on mean horizontal offsets, *X^2^*
_Friedman_ (2) = 14.10, *p* < 0.001, *W_Kendall_* = 0.23. Specifically, pairwise comparisons adjusted with Bonferroni correction showed that the mean horizontal offset was less prominent in the initial experimental stage (*M* = −23.68, SD = 49.67) compared to the second (*M* = −48.89, SD = 46.35, *p* = 0.002) and third experimental stages (*M* = −54.10, SD = 37.27, *p* = 0.001).

Saccadic reaction times in response to peripheral targets and dwell times over the face regions were extracted following individual offset correction. At this stage, trial validity was assessed. Only trials in which the infant was (1) looking at the centre of the screen at the offset of the attention grabber and (2) orienting towards the peripheral target between 100 ms and 5 s from its onset without gazing outside the screen were considered valid. Five infants who ended up with less than 20% valid data were excluded from further analysis. Out of a total 807 trials pooled across participants, 444 trials (55.02%) were valid and analysed further. Eighteen trials were also excluded as outliers. Results showed that infant participants detected the peripheral target after an average of 1269 ms (SD = 581 ms). At this time, the moving face target was located at 55° eccentricity. Dwell times over the face area were on average 2568 ms per trial (SD = 1505 ms).

## 4. Discussion

The goal in the current investigation was to generate a remote eye-tracking procedure that could successfully address some of the most relevant challenges that researchers face when studying infant participants. First, this method can accommodate head movements in a wide testing environment while measuring gaze in response to stimuli presented at 60° either side from the centre of a curved monitor. In addition, a simple offline calibration procedure was implemented. This not only improved data quality but it was also suitable for non-standard tracking areas and infant participants who cannot follow instructions.

In this study, data loss due to head movements was limited because multiple cameras were used and other cameras could take over when one camera could not acquire data, resulting in the possibility of accommodating head movements across the entire curved monitor, covering a FOV of 126°. Robustness was particularly good for a sample of infant participants. In fact, the current average data loss of 23.97% is not too far from the percentage of data loss reported for adults tested under optimal laboratory conditions, which can reach 20% [3] (pp. 166–167). The proportion of data loss included blinks and the system failing to record data for technical difficulties or for systematic infant behaviours, such as covering their eyes or orienting towards the caregiver. Data loss in the present work showed an improvement compared to the 40% data loss reported by Pratesi and collaborators [14], who also used this eye-tracking system with an infant sample. It could be that in the current study the target stimuli presentation was triggered by the infant looking at the central attention getter and, thus, the visual presentation progressed when the infant participant looked at the screen. Not having a gaze-contingent trial presentation could possibly lead to more significant data loss.

The four eye-tracking cameras adopted in the current study kept tracking both eye and head movements within the whole testing environment, covering a large visual area of 126° (although this can potentially be increased to 360° with the use of eight cameras, as reported by the manufacturer). In the present study, the focus was to measure saccadic reaction times and dwell times across a wide horizontal area but more locations, including a bigger vertical area, could be introduced by adjusting camera placement. During the entire recording, about 28% of gaze data were localised outside the wide screen but, notably, those data were within the working space of our set-up and were still recorded. This value could vary depending on how engaging the infant finds the experimental procedure. In the present study, we found out that infants’ attention was sometimes captured by the flash producers below the screen. It seems relevant for future infant studies to carefully consider the position of the flash producers.

Apart from being able to track eye gaze outside a limited headbox, the tracking system also allowed us to monitor the distance between the infants’ head from the screen throughout the testing session. Although parents were instructed to keep infants on their lap at a set distance, there was a high degree of variation in infants’ head distance during the procedure. Importantly, this method can record gaze data across a range of head–monitor distances and accommodate this expected variation in terms of distance due to the nature of this population. Normally, the head component of infant orienting behaviour is not considered in standard eye-tracking procedures. The ability to investigate infants’ orienting behaviour in a wider visual field, where head positions are less restrained, is essential in the aim to transition from strict laboratory-controlled environments to more naturalistic settings that best represent our everyday experiences. The present work provides some preliminary insights into infant information detection across a wide horizontal visual environment in which the contributions of both the eye and head components are necessary to successfully detect the target. To the best of our knowledge, a similar Smart Eye eye-tracking system has been used only once in infancy, but with a very small sample size [14]. In the present work, a new software version, specifically designed to recognise the facial proportions and anatomical features typical of infants, was adopted, and its performance was enhanced by implementing an offline calibration procedure.

Here, data quality was also considered and, in particular, spatial accuracy. Offline data inspection revealed a systematic top-right error in the recorded gaze location compared to the true gaze location (i.e., actual target position). This top-right shift was noticeable in the vast majority of individual data. Overall, the offset direction did not change as the experiment progressed in time. The vertical offset size was stable across experimental stages, while the horizontal offset size showed a slight deterioration at the beginning of the study, only to then remain stable. The average error size that was found in the present work is comparable to past findings in infancy research, which used both an initial gaze calibration and a calibration verification procedure [25,38]. However, there was some variability between each individual’s average error. Notably, the systematic gaze position error that was reported here may not be detected in standard calibration displays presented before the start of the experiment and may change following an initial online calibration. For this reason, we proposed an offline calibration procedure to correct for individual offsets and improve data quality. This method has been more widely used in recent years, especially in adult research [26,29,30,31,39,40]. It has been proposed to overcome the effect of individual factors that limit eye-tracking data quality, such as physiological features of the eye or head movements [26,41] and to account for a degradation of calibration over time [42]. In infant studies, this approach is not used as frequently as in adult research (see [24] for a study that both verified and corrected a drift in infant gaze data), even though different researchers have previously raised concerns about the accuracy of infant eye-tracking data [18,19,24,25,43]. In the current study, the implementation of an offline calibration procedure has been essential in making sure that a systematic offset was discovered and corrected, and also in overcoming the difficulty of calibrating a non-standard wide tracking environment. Further, standard gaze calibration procedures may not always be exact in developmental studies because infants do not always fixate on the required calibration points for sufficient amounts of time, and this can result in at least 1° error in spatial accuracy [1,22].

Improving spatial accuracy and estimating each individual’s offset are particularly important aspects for data interpretation, especially in some experimental designs. As outlined by Aslin [1] and Dalrymple et al. [18], spatial accuracy is extremely relevant if the eye-tracking paradigm relies on whether or not the subject looks at an area of interest (AOI). In fact, poor spatial accuracy can result in gaze points erroneously being recorded as falling outside or inside an AOI, particularly if the AOI is small and/or in close proximity to another. Furthermore, for experimental designs in which different age groups or populations are compared, discrepancies in data quality can potentially produce false differences in the outcome measures, therefore leading to erroneous interpretations [19,23].

One additional advantage of implementing an offline calibration procedure is that data are evaluated and corrected throughout the whole experiment, whereas standard calibration only occurs at the beginning of the session and is very rarely repeated during the experiment. Offline calibration therefore enables more accurate data throughout a testing session and improves the validity of eye-tracking investigations in infancy. This approach could be implemented across eye-tracking systems and is not dependent on one particular hardware system (see [24,25] for similar examples with Tobii systems). Overall, an offline calibration procedure should be adapted according to the experimental design and observed data quality. In the present work, we took advantage of the time intervals during which the visual stimuli already included in the experimental procedure were stable on screen. This allowed for six calibration points, spanning across different spatial locations and experimental stages. Whether or not all the stable visual stimuli on screen can be used as calibration points depends on data robustness and on whether gaze data are available for a sufficient duration when the target is on screen. When participants are likely not to attend to the stimuli on screen or when data quality is low in terms of robustness, additional calibration targets should be included. Notably, stable visual stimuli may not be required in every experimental procedure but should be incorporated specifically if an offline calibration procedure is planned. In this case, the calibration stimuli should ideally cover the entire tracked area. Including an increased number of stable visual stimuli than those available in the current procedure may enable future investigations to understand the ideal number of offline calibration samples that are needed to obtain the most reliable correction. Overall, even if an offline calibration step is not implemented to correct the offset, we strongly believe it is important to report not only the manufacturer’s accuracy data (albeit usually based on adult data under optimal testing conditions), but also to extract the actual data accuracy and consider its overall effect on data processing and interpretation. Downstream, data accuracy could be used as a potential parameter to exclude individual data [44], or as a guide for setting the size of the AOI [18,45].

One limitation of the current investigation is that the visual targets of the experimental procedure only appeared along the horizontal axis. For this reason, offset coordinates were collected at different eccentricities but were limited to the horizontal axis. A more accurate offline tracking procedure could include more diverse locations on screen, including along the vertical meridians and locations near the edges of the screen where spatial accuracy typically decreases. This was not possible with the setup and procedure used in the present experimental paradigm. In general, the offset that was detected in the current study was rather linear and had a consistent direction throughout the procedure. This enabled us to correct it with a simple offline calibration interface. Different offline calibration procedures may be needed if data quality is more heterogeneous in time and space. Further, although the software used in the present investigation considered the anatomy of the developing eyes and head, we did not take into account individual characteristics of our infant sample (for instance, eye colour or infant positioning during the procedure as reported by Hessels at al. [43]) that might have influenced the accuracy of this system and, more generally, data quality. More investigations are needed to identify which factors can affect data quality in wide-angle testing environments and with this specific eye-tracking system. Still, we highlighted the importance of verifying, and eventually improving, data quality parameters according to the adopted experimental procedure.

## 5. Conclusions

The current paper presents a useful tool for tracking gaze in a wide visual area without any physical constraint for the participant. This tool is suitable for infants and can accommodate high variation in head position. Additionally, the importance of considering individual spatial accuracy was highlighted and a simple interface to improve data quality was proposed. This approach is a promising methodological advance that can directly address some of the larger challenges present in infant eye tracking.

## Figures and Tables

**Figure 1 sensors-23-00972-f001:**
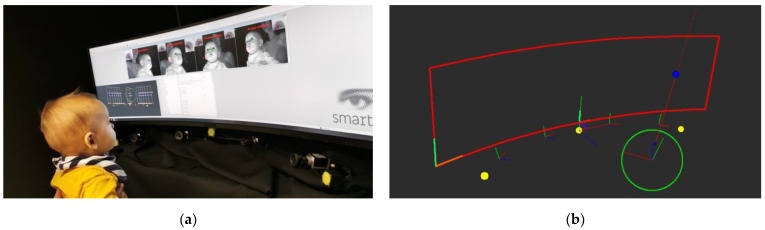
(**a**) Photographic representation of the apparatus adopted in this study. The eye-tracking system, including four cameras and three infrared flashes, was positioned below the display monitor. In this picture, the participant was ready for the system calibration step; (**b**) Image taken from the recording interface displaying the infant’s head (green circle) and their gaze position (blue dot) on the curved monitor (also defined as World Model; red area) during the experimental procedure. The yellow dots represent the three infrared flashes, and the four cameras were located in between those.

**Figure 2 sensors-23-00972-f002:**
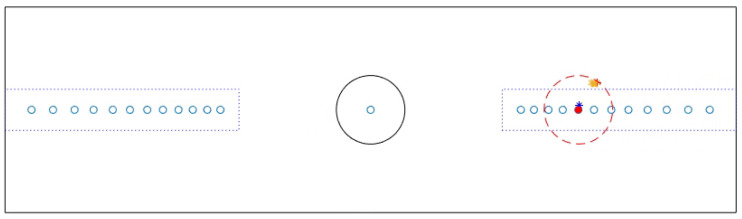
The interface implemented to review the recordings and to collect offline calibration points at different stages of the experiment. This image depicts a frame of a 2D representation of the testing display with the position of the visual target currently on screen (red dot), the initial estimated gaze points (yellow stars), and the new calibrated gaze coordinates (blue star).

**Figure 3 sensors-23-00972-f003:**
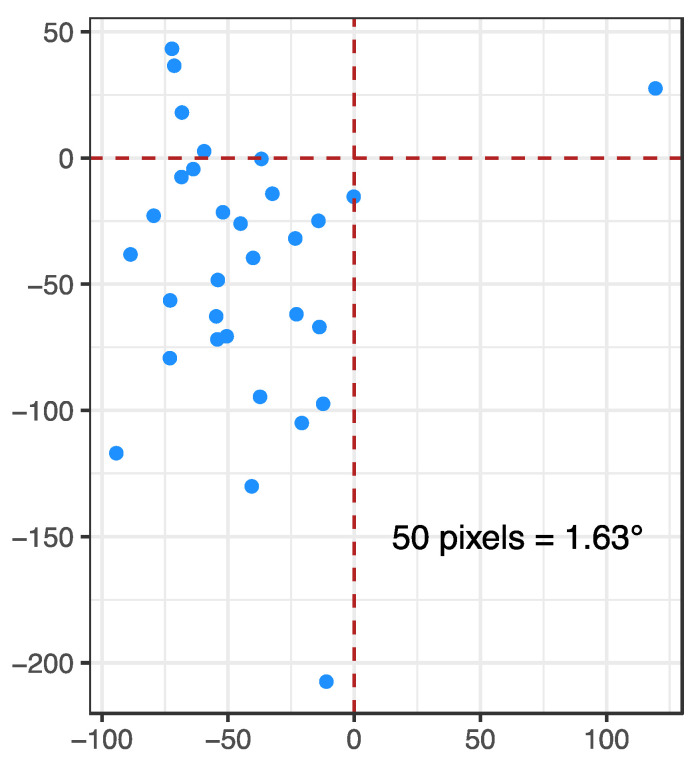
Mean individual gaze offset from the position of the target object, located at the intersection of the red dotted lines.

## Data Availability

The data presented in this study are available on request from the corresponding author.

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
