# Peer review of "Offline Calibration for Infant Gaze and Head Tracking across a Wide Horizontal Visual Field"

_sensors, 2023, doi:10.3390/s23020972_

Round 1

Reviewer 1 Report

The article presents a methodological advancement - the use of four cameras in remote eye-tracking to track infants’ eye gaze and head across a wide angle of visual field (~120 degrees). The study has a decent sample size of 9-month-olds and offers a fairly detailed and clear account of running both online and later, off-line calibration. Overall, this is a useful paper that will help many teams develop remote eye-tracking with much better accuracy for more naturalistic, wide-angle displays (or even real-world presentations). 

I have mostly some requests for clarifications and a few suggestions for expanding few elements of the paper.

Major comments:

  1. The paper claims to develop a method for more efficient tracking of the head, in addition to eye gaze. Unfortunately, the Results section does not offer any data on this aspect. I would ask the Authors to either add data on how the head tracking was done (proportion of samples, some information how the head tracking actually works in practice, ie which elements are tracked; how was validation of tracking done in this case?) or tone down their claims on superior tracking of the head and focus solely on eye gaze.
  2. The idea of off-line calibration, ie using a (constant?) correction to the already acquired data on the basis of semi-manually detected samples is very nice if the offset can be easily visualised, however, a lot of detail is missing in the description of this procedure to make it work for other researchers: 1) the Authors should clarify why and whether these drifts were constant throughout the recording (is there any data to back this up?); 2) I have doubts as to how reliable the data on drifts is, given that it was calculated on the basis of only 6 samples. Can the Authors run additional analysis at least for some participants to demonstrate that this is a sufficient number of samples to review and check? Estimating the reliability of this procedure is crucial to implement it in other studies, where more complex tasks could produce different pattern of gaze recordings errors.
  3. The Results section leaves one wanting more detailed presentation of the descriptives (including ranges and plots showing distribution). Also, the samples that were recorded outside the screen - please clarify - were they evenly distributed across all screen edges? Finally, the claim about head distance variability in l. 390 that the „system could still accommodate this variation and produce meaningful results” is not backed up by any data. Please add data to justify it.

Minor comments:

l. 32 - Im missing some references in relation to pupil dilation studies, especially with infants

l. 116 - the Wass et al. 2014 paper does not discuss eye-tracking of wide-angle displays - this should be clarified 

l. 184 - it would be very. useful if the Authors could elaborate on how the changing anatomy of the eyes and the head affects eye-tracking data acquisition across infancy

l. 190 - it is crucial that the Authors provide much more details on the exact set up of their four cameras (their exact positions and location in relation to the screen in three dimensions) and light sources. Also, how are correct angles of the cameras set up?

l. 222 - the Authors have not tested such a wide range of distances from the screen, so the range od 30-300 cm is not backed up by any data. Also, it is crucial that the Authors discuss the seating issues - would this set up work equally well for infants seated in a car seat as well as on the parent’s lap?

l. 232 - its not clear how the calibration procedure works in terms of time - can you add more detail on how this work in practice for each calibration point? What if the infant moves the head in the middle of calibration procedure? 

l. 239 - clarify what the parent’s were doing exactly to maintain the infant’s head within the headbox

l. 246 - unclear - „that each camera could detect the chessboard” - presumably the eyes on the chessboard?

l. 261 onwards - please state full details of stimuli sizes and their positions, including the attention getter

l. 266 onwards - its not clear what happened if the infant did not fixate the central attention getter within the 900 ms  - was it automatically presented again (with a period of break?)

l. 296 - explain what are „World Intersections”

l. 301 - explain how the gaze positions were recorded if they were outside the screen (how was that data represented?)

l. 411 - its not clear whether this is trials pooled across participants

l. 462 - can you estimate the size of the head box? The statement that its „large” is not very specific

Reviewer 2 Report

The paper is clear and straightforward. There are just few aspects, which deserve further attention.
- The data analysis is very poor: in order to be more effective, Authors are required to add statistics to their evaluations. For example, is it possible to evaluate whether there is a different behavior while looking to the center and laterally? How much was the rate of improvement from the first target to the last one? Etc... Try to provide as much reference data as possible, to support the claims.
- Would it be possible to model an algorithm that, by inserting the data collected by a user, could be able to correct automatically the gaze offset in x and y?
- In the discussion, very few references to past literature and models were present. It would be relevant to clearly identify methodological work about eye-tracking research so that, for example, the general "factors to be determined" (as stated in the limitations) would be already determined.
